# Endothelial ADAM17 Expression in the Progression of Kidney Injury in an Obese Mouse Model of Pre-Diabetes

**DOI:** 10.3390/ijms23010221

**Published:** 2021-12-25

**Authors:** Vanesa Palau, Josué Jarrín, Sofia Villanueva, David Benito, Eva Márquez, Eva Rodríguez, María José Soler, Anna Oliveras, Javier Gimeno, Laia Sans, Marta Crespo, Julio Pascual, Clara Barrios, Marta Riera

**Affiliations:** 1Department of Nephrology, Hospital del Mar-Institut Hospital del Mar d’Investigacions Mèdiques, 08003 Barcelona, Spain; vpalau@imim.es (V.P.); JavierJCarriel@outlook.com (J.J.); villanuevas2c@gmail.com (S.V.); david.benito.guasch@gmail.com (D.B.); eva.marquez.mosquera@psmar.cat (E.M.); erodriguezg@psmar.cat (E.R.); aoliveras@psmar.cat (A.O.); lsans@psmar.cat (L.S.); MCrespo@parcdesalutmar.cat (M.C.); julpascual@gmail.com (J.P.); 2Nephrology Research Group, Vall d’Hebron Research Institute (VHIR) and Department of Nephrology, Hospital Universitari Vall d’Hebron, Universitat Autònoma de Barcelona, 08035 Barcelona, Spain; mjsoler01@gmail.com; 3Department of Pathology, Hospital del Mar, 08003 Barcelona, Spain; JGimenoBeltran@parcdesalutmar.cat

**Keywords:** ADAM17, obesity, pre-diabetes, endothelial cells

## Abstract

Disintegrin and metalloproteinase domain 17 (ADAM17) activates inflammatory and fibrotic processes through the shedding of various molecules such as Tumor Necrosis Factor-α (TNF-α) or Transforming Growht Factor-α (TGF-α). There is a well-recognised link between TNF-α, obesity, inflammation, and diabetes. In physiological situations, ADAM17 is expressed mainly in the distal tubular cell while, in renal damage, its expression increases throughout the kidney including the endothelium. The aim of this study was to characterize, for the first time, an experimental mouse model fed a high-fat diet (HFD) with a specific deletion of *Adam17* in endothelial cells and to analyse the effects on different renal structures. Endothelial *Adam17* knockout male mice and their controls were fed a high-fat diet, to induce obesity, or standard rodent chow, for 22 weeks. Glucose tolerance, urinary albumin-to-creatinine ratio, renal histology, macrophage infiltration, and galectin-3 levels were evaluated. Results showed that obese mice presented higher blood glucose levels, dysregulated glucose homeostasis, and higher body weight compared to control mice. In addition, obese wild-type mice presented an increased albumin-to-creatinine ratio; greater glomerular size and mesangial matrix expansion; and tubular fibrosis with increased galectin-3 expression. *Adam17* deletion decreased the albumin-to-creatinine ratio, glomerular mesangial index, and tubular galectin-3 expression. Moreover, macrophage infiltration in the glomeruli of obese *Adam17* knockout mice was reduced as compared to obese wild-type mice. In conclusion, the expression of ADAM17 in endothelial cells impacted renal inflammation, modulating the renal function and histology in an obese pre-diabetic mouse model.

## 1. Introduction

The incidence of obesity is increasing exponentially in the developed countries, contributing to the increasing prevalence of various pathological conditions of obesity-related metabolic disorders [1]. Obesity enhances the risk factors for chronic kidney disease (CKD), such as type 2 diabetes and hypertension [2].

In an obese state, adipocyte hypertrophy disturbs the balance of adipose tissue-derived cytokines and adipokines, leading to a proinflammatory state with subsequent adipocyte dysfunction [3]. This is also associated with increased secretion of proinflammatory cytokines, causing a typical infiltration of activated macrophages that further promotes the inflammatory process [4]. These alterations can lead to the development of systemic inflammation, oxidative stress, abnormal lipid metabolism, activation of the renin–angiotensin–aldosterone system, increased production of insulin and insulin resistance, peripheral complications, vascular endothelial cell dysfunction, and atherosclerosis [1,5].

Disintegrin and metalloproteinase domain 17 (ADAM17) is a type I transmembrane protein that was initially described to specifically cleave the pro-inflammatory cytokine tumour necrosis factor *α* (TNF-*α*) [6]. Currently, it is known that ADAM17 can also release the ectodomains of a diverse variety of membrane-anchored cytokines, cell adhesion molecules, receptors, ligands, and enzymes [7,8,9,10,11]. Most of them are associated with disorders, such as obesity, diabetes, cardiovascular disease, autoimmune diseases, hypertension, and cancer [8,9,12]. Several reports support the relationship between ADAM17 and insulin resistance. In high-fat diet (HFD) mice, overexpression of ADAM17 increased TNF-*α*, macrophage infiltration, fibrosis, and inflammatory cytokines in the adipose tissue [13]. The pro-inflammatory state induced by ADAM17 led to glucose intolerance and insulin resistance in HFD mice [14]. In contrast, *Adam17* deletion or pharmacological inhibition improved glucose tolerance and insulin sensitivity in HFD mice [15,16,17].

In the kidney, high glucose increases ADAM17 expression in different renal cells, such glomerular endothelial cells, mesangial cells, proximal tubular epithelial cells, and podocytes [18,19,20,21]. Understanding the molecular basis of vascular endothelial dysfunction is needed to develop more strategies to reduce the incidence of cardiovascular diseases associated with obesity. In the present study we aimed to evaluate whether conditional *Adam17* deletion in endothelial cells is able to protect different renal structures from the deleterious effects of obesity-mediated inflammation.

## 2. Results

### 2.1. Endothelial Adam17 Deletion Modifies Blood Glucose Levels and Renal Function in Obese Mice

To investigate the impact of endothelial *Adam17* deletion on the kidneys of obese mice, male animals fed with HFD were studied for 22 weeks. As shown in Table 1, the obese groups weighted more as compared to standard diet (SD)-fed mice at the end of the study. Wild-type obese mice exhibited increased kidney weight as compared to wild-type control mice. Notably, no differences were observed between knockout groups.

As depicted in Table 1, fasting blood glucose levels were higher in all obese mice as compared to SD-fed mice, but the values did not exceed 250 mg/dL. Therefore, HFD-mice were considered prediabetic, as described before [22]. Urinary albumin was calculated by the albumin-to-creatinine ratio (ACR). As shown in Table 1, only wild-type obese mice had a significant increase in ACR as compared to SD mice. Interestingly, obese knockout mice presented a significant decrease in ACR values as compared to obese wild-type mice.

### 2.2. Glucose Tolerance was Modulated by Adam17 Deletion in Endothelial Cells

To analyse the impacts of HFD and endothelial *Adam17* deletion, glucose tolerance was measured by performing intraperitoneal glucose tolerance tests (IPGTT) on all groups (Figure 1).

Only obese wild-type mice demonstrated significant glucose intolerance compared to control mice at all time points after glucose bolus injection. Interestingly, obese knockout mice did not show statistical differences in glucose intolerance post-bolus injection as compared to knockout mice fed a SD. Moreover, a tendency toward decrease glucose tolerance was observed in obese knockout mice as compared to standard chow groups without reaching statistical significance.

In line with the previous results, the area under the curve (AUC) value was only increased in obese wild-type mice fed an HFD compared to control group (Table 2). Interestingly, *Adam17* deletion in endothelial cells prevented obese mice from increased AUC values compared to controls.

### 2.3. Adam17 Deletion in Endothelial Cells Protects Obese Mice from Glomerular Alterations

Histological periodic acid-Schiff (PAS)-stained kidney samples revealed a significant increase in glomerular tuft area and mesangial matrix expansion, which was reflected in the mesangial index (mesangial area/glomerular tuft area) of obese wild-type mice. This effect of obesity was not observed in knockout mice. Interestingly, obese knockout mice showed a significant reduction in mesangial index as compared to obese wild-type mice (Figure 2A).

The number of podocytes per glomerulus was significantly decreased in obese wild-type mice in comparison with controls. However, this decrease in podocyte number was not observed in obese mice with endothelial *Adam17* deletion as compared with knockout mice fed a SD (Figure 2B).

### 2.4. Endothelial Adam17 Promotes Renal Macrophage Infiltration in Obese Mice

To evaluate whether deletion of *Adam17* in endothelial cells protects from renal inflammation, F4/80 immunohistochemistry was performed as a marker of macrophage infiltration.

As shown in Figure 3, obese wild-type mice presented increased macrophage infiltration in the interstitial compartment. Interestingly, *Adam17* deletion from endothelial cells was able to decrease the number of macrophages in the renal interstitia of obese mice as compared with obese wild-type mice.

### 2.5. Renal Galectin-3 Expression Is Modified by Endothelial Adam17 Deletion

As depicted in Figure 4A, galectin-3 expression was found upregulated in obese wild-type mice as compared to controls. In contrast, this effect of HFD was lost in knockout animals. 

Immunostaining revealed increased expression of galectin-3 in the cortical tubules of all obese mice. Interestingly, mice with endothelial *Adam17* deletion presented a tendency of decreased galectin-3 in cortical tubules (Figure 4B).

Sirius Red staining for collagen type I and type III localization was also performed. Increased collagen accumulation was found in the glomeruli of obese mice. However, no differences were observed between wild-type and knockout animals (data not shown).

## 3. Discussion

High glucose induces increased ADAM17 protein expression in different renal cells, namely, glomerular endothelial cells, mesangial cells, proximal tubular epithelial cells, and podocytes [18,19,20,21], contributing to renal inflammation and fibrosis [23,24,25]. Since endothelium dysfunction has been posited to play an important role in the pathogenesis of diabetic nephropathy, we aimed to study the effect of conditional endothelial *Adam17* deletion on an obese pre-diabetic mouse model.

HFD-feeding in mice is known to induce various systemic alterations, including obesity, hyperglycaemia, and abnormal lipid profile, which are similar to those observed in patients with metabolic syndrome [26]. Several studies have reported that mice on a HFD show several types of renal injury, including the core features of metabolic syndrome, moderate albuminuria, and glomerular lesions [27,28,29,30]. Our animal model satisfactorily demonstrated the key features of type 2 pre-diabetes, including a > 40% increase in body weight, higher blood glucose levels, an increased albumin-to-creatine ratio, and increased renal hypertrophy.

Focusing on glucose homeostasis, obese animals with *Adam17* deletion on endothelial cells did not present increased glucose intolerance as compared with controls after 60 min of bolus injection. Probably this mild protection against glucose intolerance can be explained by the modulation of Sodium-Glucose Cotransporter-2 (SGLT2) after *Adam17* deletion. As we have recently described [31], *Adam17* deletion may decrease SGLT2 activity, preventing HFD animals from proximal tubular glucose reabsorption by favouring its urinary excretion. HFD-mice with *Adam17* deleted in endothelial cells presented decreased SGLT2 expression. These results favour our hypothesis that less glucose reabsorption could lead to better glucose tolerance (see Appendix A).

In humans, moderate albuminuria is a biomarker and major risk factor for progressive renal function decline in diabetes, and is thought to be the first step toward severe albuminuria progression and renal failure [32,33,34]. Thus, reduction of albuminuria is a major target for renoprotective therapy. Huang MJ et al. found that inflammatory and oxidative stress markers such as interleukin-2, interleukin-6, and superoxide dismutase had strong correlations with ACR in CKD patients. They hypothesized that these factors may mediate the association between ACR and endothelial dysfunction, suggesting that inflammation and oxidative stress markers change as ACR increases in CKD patients [35]. Previous studies in diabetic animals demonstrated a slight reduction in ACR levels after treatment with paricalcitol [36,37], an analogue of the active form of vitamin D with ADAM17 inhibition properties [24,38]. Accordingly, in our study, obese mice with *Adam17* deleted in endothelial cells exhibited a decreased ACR ratio as compared to obese wild-type mice. These results suggest that ADAM17 in endothelial cells promotes increases in inflammatory and oxidative stress molecules that favour glomerular injury, leading to the increase on ACR. Therefore, decreasing ADAM17 at the endothelial level may have a renoprotective effect and could be a possible translational strategy for patients with incipient type 2 diabetes. 

It is known that vascular dysfunction may eventually lead to a decline in renal function and the development of glomerulosclerosis and tubulointerstitial fibrosis. Endothelial dysfunction also diminishes the antiatherogenic ability of endothelial cells, which may also contribute to abnormal renal function [39]. Obesity-associated renal injury is characterized by glomerulomegaly, mesangial expansion, and podocytopenia leading to focal glomerulosclerosis [40]. As expected, our obese wild-type mice presented glomerular alterations that included an elevated mesangial index together with a decreased podocyte number. Ziyadeh FN et al. proposed that increased adiposity triggers the release of adipokines into the circulation, causing renal injury via production of reactive oxygen species. High blood glucose levels also induce vasoactive hormonal pathways—including the renin–angiotensin system—that lead to the activation of protein kinase C (PKC), MAP kinase (MAPK), nuclear factor-*κ*B (NF-*κ*B), and transforming growth factor-beta (TGF-*β*) signalling pathways. TGF-*β* has been suggested a key player in promoting podocyte apoptosis, mesangial cell proliferation, and extracellular matrix synthesis, cellular events that are important in the development of obesity-associated glomerular injury [41]. Several studies suggested that the upregulation of sterol-regulatory element binding proteins (SREBP-1 and 2) in hyperglycaemia and hyperlipidaemia promotes TGF-*β* signalling, leading to glomerular injury in terms of podocyte apoptosis, mesangial cell proliferation, and cytokine synthesis [42,43,44,45]. Interestingly, in the present study we demonstrated that *Adam17* deletion in endothelial cells prevented obese mice from developing glomerular lesions by decreasing mesangial matrix expansion and slowing down glomerular hypertrophy and podocyte loss. In this sense, we hypothesized that *Adam17* deletion in endothelium may protect obese pre-diabetic mice from glomerular lesions by decreasing glomerular inflammation and TGF-*β* signalling. 

Obesity exerts a low-grade inflammation state associated with macrophage infiltration into the adipose tissue and the kidney. The infiltrating macrophages become a source of pro-inflammatory cytokines, including TNF-*α*, interleukin-6, and MCP-1 [46]. Several studies have reported increased macrophage infiltration in the glomerular and interstitial compartments of obese mice [47,48,49]. Accordingly, we observed elevated numbers of macrophages in the interstitia of obese mice. Noteworthy is that *Adam17* deletion in endothelial cells decreased macrophage infiltration in obese mice. Relatedly, we previously demonstrated decreased cortical inflammation and interstitial macrophage infiltration in type 1 diabetic mice with *Adam17* deletion in endothelial cells [50]. Moreover, Awad AS et al. demonstrated that TNF-*α* inhibition is also linked to decreased macrophage infiltration in type 1 diabetic mice [51]. These results suggested that endothelial ADAM17 contributes to the release of inflammatory molecules, which triggers macrophage infiltration. 

Galectin-3 is a *β*-galactoside-binding lectin that has a regulatory role in inflammatory and fibrotic processes [52]. Circulating galectin-3 has been suggested as a potential biomarker for diabetic nephropathy progression and cardiovascular disease prognosis linked to myocardial fibrosis, tissue remodelling, and heart failure development [53,54,55]. In patients with altered renal function, increased plasma levels of galectin-3 have been associated with higher risk of renal function decline, incident chronic kidney disease, and renal failure [52]. Galectin-3 is involved in many processes during acute inflammatory responses, including neutrophil activation and adhesion and chemoattraction of macrophages [56,57]. Lu HY et al. demonstrated that galectin-3 is responsible for macrophage activation by inducing monocyte-macrophage differentiation and amplifying inflammation [56]. Moreover, several studies have postulated that macrophages are the major source of galectin-3 driving renal fibrosis through myofibroblast activation [58,59]. In our pre-diabetic mouse model, galectin-3 expression was increased in obese wild-type animals, and *Adam17* deletion on endothelial cells decreased galectin-3 levels in the cortices of obese mice. These results suggest that *Adam17* deletion in endothelial cells prevents the recruitment of macrophages into the kidney, and in consequence, increases renal galectin-3 levels in obese pre-diabetic mice.

## 4. Materials and Methods

### 4.1. Animal Experiments

Experiments were performed on wild-type and endothelial *Adam17KO* male mice on a C57BL/6 background. Mice were housed in ventilated cages with full access to chow and water. *Adam17* was conditionally deleted in endothelial cells, as previously described by our group [50]. Briefly, the generation of specific endothelial knockout mice occurred by crossing Adam17^flox/flox^ mice (kindly provided by Dr. Raines, Washington University, St. Louis, MO, USA) [60] and tamoxifen-inducible platelet-derived growth factor (Pdgf)-iCreER mice (kindly provided by Dr. Fruttiger, University College London, UK) [61]. *Adam17^flox^*^/flox^ mice presented two loxP sites surrounding the exon 5 of the *Adam17* gene. After recombination, the excision of the sequence results in a frame shift producing a nonsense protein [60]. To induce *Adam17* gene recombination in endothelial cells, five intraperitoneal doses of 0.1 mg/g body weight of tamoxifen were administrated to 10-week-old mice, as previously described [50]. Wild-type (WT) mice receiving tamoxifen were used as controls.

To induce obesity, at 10 weeks of age, mice were divided randomly into two groups according to diet. Wild-type (*Adam17-WT*) and knockout (*Adam17-KO*) mice were fed a high fat diet (HFD, 60.3% fat; TD.06414, Envigo, Indianapolis, IN, USA) or standard rodent chow (SD) (controls; 7.4% fat; 801151, SDS) for 22 weeks (Appendix A). Body weight and fasting blood glucose (BG) from the caudal vein were measured every two weeks until the end of the follow-up. Mice were considered diabetic when BG > 250 mg/dL.

At the end of the follow-up, mice were sacrificed by terminal surgery as previously described [62]. Blood was extracted by cardiac puncture and serum was obtained by centrifugation at 6000× *g* for 10 min. Mice were perfused with cold PBS prior to kidney removal and weighting. Half of the right kidney was maintained in 10% formalin solution for paraffin embedding. The remaining tissue was snap frozen on liquid nitrogen and kept at −80 °C for further analyses.

### 4.2. Glucose Tolerance Test

An intraperitoneal glucose tolerance test (IPGTT) was performed 3 days before the end of the study. Mice were fasted for 6 h prior to the test as previously described [63]. Briefly, 2 g/kg of D-glucose (Sigma-Aldrich, St. Louis, MO, USA) were administrated intraperitoneally and blood glucose levels were recorded after 15, 30, and 60 min of bolus injection with an Accu-Chek Compact glucometer (Roche, Basel, Switzerland).

### 4.3. Urinary Albumin-to-Creatinine Ratio

Urinary albumin excretion (UAE) was determined by using the albumin-to-creatinine (ACR) ratio of morning spot urine collections obtained on the last week of the follow-up through abdominal massage. Urinary albumin levels were measured by ELISA kit (Albuwell M, Exocell, Philadelphia, PA, USA). Creatinine levels were measured by colorimetric assay (Creatinine Companion, Exocell, Philadelphia, PA, USA). Albumin-to-creatinine ratio was calculated and expressed as µg Alb/mg Crea [24].

### 4.4. Immunohistochemistry on Paraffined-Embedded Tissue

Paraffin-embedded tissues were cut into 3 µm-sections, deparaffined in xylene, and rehydrated through graded alcohols. Sections were stained with periodic acid-Schiff (PAS) for glomerular area and mesangial matrix expansion measurements as previously reported [64]. Twenty microphotographs of glomeruli were taken at 400× magnification for each animal. Image J software was used to analyse glomerular areas.

Immunohistochemistry for podocyte marker Wilms Tumor 1 (WT-1), galectin-3 (Gal-3), and macrophage marker F4/80 was also performed with sections of paraffin-embedded tissue. Antigen retrieval was carried out with 0.01 M sodium citrate buffer pH6 by heating in a pressure cooker. Endogenous peroxidase was blocked by 3% H_2_O_2_ in TBS1X incubation for 15 min. Non-specific interactions blocking was done by 1% BSA and 3% goat serum for 1 h. Sections were then incubated with anti-WT-1 antibody (1:1000; sc192, Santa Cruz Biotechnology, Dallas, TX, USA), anti-galectin-3 antibody (1:1000; 126701, Biolegend, San Diego, CA, USA), or anti-F4/80 antibody (1:500; 400501, Biolegend, San Diego, CA, USA) at 4 °C overnight. After washing, the slides were incubated with HRP-conjugated anti-rabbit IgG, anti-mouse IgG, or anti-rat IgG for 1 h at room temperature.

Binding of the antibodies was detected by oxidation of DAB using the Liquid DAB + Substrate Chromogen System (Dako, Santa Clara, CA, USA). Samples were counterstained with haematoxylin and dehydrated through graded alcohols and preserved with DPX mounting media (Sigma-Aldrich, St. Louis, MO, USA). Twenty microphotographs of glomeruli immunostained with anti-WT-1 were taken at 400× magnification. Representative microphotographs of renal cortex immunostained with anti-Gal-3 or anti-F4/80 were taken at 100× and 200× magnification, respectively.

### 4.5. Western Blot

Kidney cortical tissue was prepared for immunoblot analysis with antibodies against phosphorylated and total Drp1. Kidney cortex samples were homogenized in extraction buffer containing 50 mM HEPES, pH7.4, 150 mM NaCl, 0.5% Triton X-100, 0.025 mM ZnCl2, 0.1 mM Pefabloc SC Plus (Roche, Basel, Switzerland), EDTA-free protease inhibitor cocktail tablet (Roche, Basel, Switzerland), and phosphatase inhibitor cocktail (Sigma-Aldrich, St. Louis, MO, USA). Protein concentration was determined using the Micro BCA Protein Assay Kit (ThermoFisher Scientific, Waltham, MA, USA).

Western blot was performed by separating 15 µg of total protein in 7% SDS-polyacrylamide gels and transferring it onto PVDF membranes (Immobilon-P Millipore, Burlington, MA, USA). Membranes were incubated in skimmed milk blocking solution (5%) for 1 h and incubated overnight at 4 °C with galectin-3 (1:500; 126701, Biolegend, San Diego, CA, USA) in 2.5% skimmed milk. HRP-conjugated anti-mouse IgG antibody was used as secondary antibody. *β*-Actin antibody (1:20,000; BS1003, Bioworld, Dublin, Ireland) was used as loading control.

Proteins were detected in films (AGFA CURIX, Mortsel, Belgium) after 3-min incubation with Clarity Western ECL Substrate (Bio-Rad, Hercules, CA, USA). Protein bands were quantified by densitometry with the ImageJ software.

### 4.6. Statistical Analyses

Statistical analyses between groups were performed by one-way ANOVA test (SPSS 22.0 software, IBM, Armonk, NY, USA). Non-parametric Kruskal–Wallis tests were performed between groups. Non-parametric Mann–Whitney tests were used for group-to-group comparisons. Data was expressed as mean ± SD. Significance was defined as *p* < 0.05.

## 5. Conclusions

Our study suggested that cell specific *Adam17* deletion in endothelial cells might have protective effects in obese pre-diabetic mice. We have demonstrated that *Adam17* deletion in endothelial cells contributes to protect glomeruli from structural alterations mainly induced by high-fat diet. Concretely, *Adam17* deletion in endothelial cells ameliorates albuminuria, glomerular hypertrophy, mesangial expansion, podocyte loss and macrophage infiltration. Additionally, ADAM17 modulates galectin-3 expression levels and affects the distribution of galectin-3 through the renal cortex. Our results propose ADAM17 as a potential therapeutic target and opens an interesting line of research as obesity and type 2 diabetes are increasing worldwide.

## Figures and Tables

**Figure 1 ijms-23-00221-f001:**
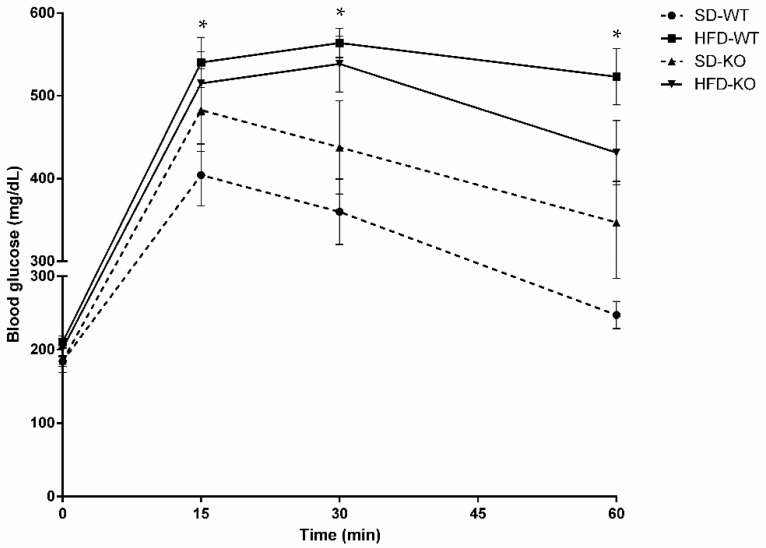
Intraperitoneal glucose tolerance tests performed at 32 weeks of age in male mice. Blood glucose levels were measured 0, 15, 30, and 60 min post glucose bolus injection (*n* = 8). Data are expressed as mean ± SEM. Abbreviations: SD, standard diet; HFD, high fat diet; WT, wild-type; KO, knockout. * *p* < 0.05 WT-HFD vs. WT-SD.

**Figure 2 ijms-23-00221-f002:**
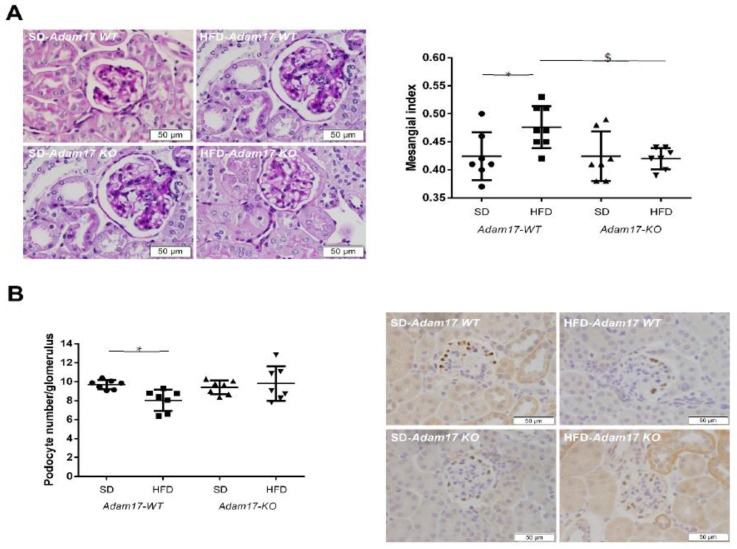
Influences of HFD and endothelial *Adam17* deletion on renal structures. (**A**) Mesangial index values (*n* = 7). Representative images depicting PAS staining from all experimental groups. 400× magnification, scale bar 50 μm. (**B**) Glomerular podocyte number represented as the number of positive cells per glomerulus after WT-1 immunostaining (*n* = 7). Representative images depicting WT-1 immunostaining from all experimental groups. 400× magnification, scale bar 50 μm. Data are expressed as mean ± SD. Abbreviations: SD, standard diet; HFD, high fat diet; Adam17WT, wild-type; Adam17KO, knockout. * *p* < 0.05 Adam17WT-HFD vs. Adam17WT-SD; $ *p* < 0.05 Adam17KO-HFD vs. Adam17WT-HFD.

**Figure 3 ijms-23-00221-f003:**
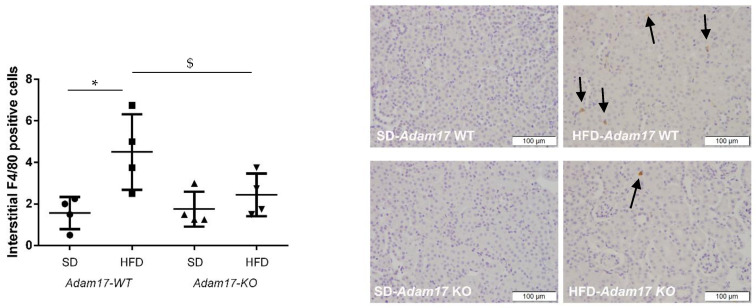
Influences of HFD and endothelial *Adam17* deletion on renal macrophage infiltration. Quantification and representative cortical images of F4/80 immunostaining (*n* = 4). Black arrows showed positive F4/80 cells. 200× magnification, scale bar 100 μm. Data are expressed as mean ± SD. Abbreviations: SD, standard diet; HFD, high fat diet; Adam17WT, wild-type; Adam17KO, knockout. * *p* < 0.05 Adam17WT-HFD vs. Adam17WT-SD; $ *p* < 0.05 Adam17KO-HFD vs. Adam17WT-HFD.

**Figure 4 ijms-23-00221-f004:**
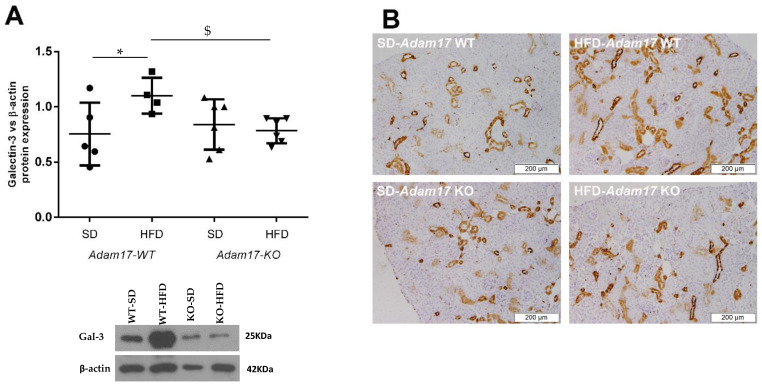
Influences of HFD and endothelial *Adam17* deletion on galectin-3 expression. (**A**) Galectin-3 expression measured by Western blot (*n* = 4–6). (**B**) Representative images of galectin-3 immunostaining. 100× magnification, scale bar 200 μm. Data are expressed as mean ± SD. Abbreviations: SD, standard diet; HFD, high fat diet; Adam17WT, wild-type; Adam17KO, knockout. * *p* < 0.05 WT-HFD vs. WT-SD; $ *p* < 0.05 KO-HFD vs. WT-HFD.

**Table 1 ijms-23-00221-t001:** Fasting blood glucose, body weight, kidney weight, and urinary albumin excretion were measured after 22 weeks of follow-up (*n* = 8). Values are expressed as mean ± SD. Abbreviations: SD, standard diet; HFD, high fat diet; Adam17WT, wild-type; Adam17KO, knockout. * *p* < 0.05 Adam17WT-HFD vs. Adam17WT-SD; # *p* < 0.05 Adam17KO-HFD vs. Adam17KO-SD; $ *p* < 0.05 Adam17KO-HFD vs. Adam17WT-HFD.

	Fasting Blood Glucose (mg/dL)	Body Weight (g)	Kidney Weight (g)	ACR (μg Alb/mg Crea)
Adam17WT-SD	185.71 ± 16.39	36.83 ± 7.79	0.32 ± 0.04	27.26 ± 11.36
Adam17WT-HFD	245.38 ± 36.29 *	50.91 ± 4.98 *	0.39 ± 0.06 *	50.43 ± 13.03 *
Adam17KO-SD	193.45 ± 17.83	40.23 ± 6.49	0.34 ± 0.05	20.16 ± 6.24
Adam17KO-HFD	235.40 ± 27.31 #	49.69 ± 6.06 #	0.38 ± 0.07	29.74 ± 9.55 $

**Table 2 ijms-23-00221-t002:** Area under the curve was calculated for each experimental group using the trapezoid method (*n* = 8). Data are expressed as mean ± SD. Abbreviations: SD, standard diet; HFD, high fat diet; Adam17WT, wild-type; Adam17KO, knockout. * *p* < 0.05 Adam17WT-HFD vs. Adam17WT-SD.

	AUC (mg/dL/min)
Adam17WT-SD	19,059.64 ± 4509.13
Adam17WT-HFD	30,221.79 ± 3366.05 *
Adam17KO-SD	23,692.5 ± 7531.78
Adam17KO-HFD	27,831 ± 3955.68

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
