# Peer review of "Endothelial ADAM17 Expression in the Progression of Kidney Injury in an Obese Mouse Model of Pre-Diabetes"

_ijms, 2021, doi:10.3390/ijms23010221_

Round 1

Reviewer 1 Report

Major;

Immunostaining or in situ should be shown to confirm endothelial cell-specific ADM17 gene deletion. It is necessary to provide details about the mouse genetic modification in Materials and Methods.

The authors have to consider why the KO mice had mild glucose intolerance, also need to consider whether the improvement in glucose intolerance reduced the renal damage in KO mice.

No significant difference can be determined for all images in Fig. 4B. The images should be replaced with more representative images at lower magnifications.

Minor;

Proximal-tubular Adam17KO male mice→Endothelial Adam17KO male mice in Line 262.

Pdgf-CreER+→Pdgfb-CreER+ in Supplementary Figure 1.

Author Response

Major;

Immunostaining or in situ should be shown to confirm endothelial cell-specific ADM17 gene deletion. It is necessary to provide details about the mouse genetic modification in Materials and Methods.

Thanks for the suggestion. We have previously demonstrated endothelial Adam17 deletion in this animal model (PMID: 34073747) by immunohistochemistry performed in frozen samples of kidneys embedded in OCT. We were not able to localize Adam17 by immunohistochemistry in the present study as we did not collect frozen kidneys embedded in OCT. The immunohistochemistry of Adam17 performed in paraffin let to a huge background that difficult the interpretation of the results.

The authors have to consider why the KO mice had mild glucose intolerance, also need to consider whether the improvement in glucose intolerance reduced the renal damage in KO mice.

We would like to thank the reviewer for his/her comments. We think that the mild glucose intolerance observed in knockout mice fed with SD is an intrinsic characteristic of the mice. Probably it is associated with the higher body weight and higher blood glucose observed in these animals. However, no statistical differences were found between wild-type and knockout mice fed with standard diet when measuring glucose intolerance.

Moreover, knockout mice with high fat diet did not present higher glucose intolerance as compared to knockout mice fed with standard diet. Probably, this improvement in glucose tolerance in the knockout mice leads to a decrease on glucose reabsorption to the circulation. The decrease on blood glucose levels attenuates the activation of inflammatory and fibrotic pathways and therefore reduces renal damage. Adam17 deletion on endothelial cells may improve glucose tolerance by decreasing SGLT2 expression. As SGLT2 is a sodium-glucose co-transporter involved in glucose reabsorption into the circulation, when SGLT2 expression is decreased, more glucose can be eliminated through the urine. We have performed an immunohistochemistry on SGLT2 to demonstrate this hypothesis. As depicted in the new Supplementary Figure 2, SGLT2 is increased in HFD-WT mice and decreases after Adam17 deletion in endothelial cells. (lines 204-212).

No significant difference can be determined for all images in Fig. 4B. The images should be replaced with more representative images at lower magnifications.

We would like to thank the reviewer for his/her suggestion. We have changed cortical Galectin-3 images with new ones more representative at lower magnifications.

 Minor;

Proximal-tubular Adam17KO male mice→Endothelial Adam17KO male mice in Line 262.

We have corrected this typo in the new version of the manuscript (see line 286).

Pdgf-CreER+→Pdgfb-CreER+ in Supplementary Figure 1.

We have modified Supplementary Figure 1 according to his/her comment.

After detailed revision of the manuscript we detected a wrong graph in figure 2B. Data description and statistical analysis were correct, but the data presentation was wrong. We apologize for the error. In the new version of the manuscript the correct graph has been added.

Reviewer 2 Report

In this study, the authors evaluated whether conditional Adam17 deletion on endothelial cells was able to protect different renal structures from the deleterious effect of obesity-mediated inflammation. The authors concluded that the expression of ADAM17 in the endothelial cells had an impact on renal inflammation modulating renal function and histology in an obese pre-diabetic mouse model.

Comments

The reviewer has some concerns as follows:

  1. Data presentation of this manuscript needs to be improved.
  2. The experimental “n” numbers in Table 2 and all Figures should be shown.
  3. In Figure 1, the standard deviation (SD) bars for all groups are lacking.
  4. The indications for “*” (HFD vs. SD) and “$” (KO vs. WT) are confusing. It can be specific.
  5. The scale bars for figures of PAS staining and IHC staining can be shown.
  6. In Figure 2A, the position for “$” label is wrong.
  7. In Figure 3A and 3B, the data for F4/80 immunostaining are really unconvincing. The numbers for F4/80 staining-positive cells are too small to convince.
  8. The data for Figure 4A and 4B are really unconvincing. In lower panel of Figure 4A, the presentation of immunoblot for Gal-3 cannot be matched with densitometrical data plot (upper panel). The immunoblot showed no difference among WT-HFD, KO-SD, and KO-HFD groups, and there was a smaller Gal-3 protein expression for WT-SD. Why there is only one blot band for WT-SD group, but two repeat blot bands for other groups? In Figure 4B, the authors do not demonstrate that the positive Gal-3 stains are really located in endothelial cells. Moreover, from these IHC images, it is hard to distinguish the changes or differences among these four groups.
  9. The authors should show evidence to demonstrate that the endothelial Adam17 expression is really deleted in KO mice.
  10. In Methods section, the descriptions for animal experiments (4.1.) are confusing. Why “proximal-tubular” Adam17KO male mice are used (line 252)? What is the source for WT and KO mice? Moreover, the authors measure the urinary ACR, but there are no descriptions for urine sampling, when and how?

Author Response

In this study, the authors evaluated whether conditional Adam17 deletion on endothelial cells was able to protect different renal structures from the deleterious effect of obesity-mediated inflammation. The authors concluded that the expression of ADAM17 in the endothelial cells had an impact on renal inflammation modulating renal function and histology in an obese pre-diabetic mouse model.

Comments

The reviewer has some concerns as follows:

Data presentation of this manuscript needs to be improved.

We would like to thank the reviewer for this comment. We fixed some figures’ characteristics according to reviewer’s comments which considerably improved data presentation.

The experimental “n” numbers in Table 2 and all Figures should be shown.

The “n” for each experimental group has been added in the figure legend of each figure of the new version of the manuscript.

In Figure 1, the standard deviation (SD) bars for all groups are lacking.

Thanks for this appreciation. SD bars have been added in the new version of the manuscript.

The indications for “*” (HFD vs. SD) and “$” (KO vs. WT) are confusing. It can be specific.

We thank the reviewer for his/her appreciation. We have better specified the indications for “*” and “$”; and added a new indication for diet comparisons in KO mice “#”. Then, in the new version of the manuscript the symbols are as follows: *p<0.05 WT-HFD vs. WT-SD; #p<0.05 KO-HFD vs. KO-SD; $p<0.05 KO-HFD vs. WT-HFD.

The scale bars for figures of PAS staining and IHC staining can be shown.

Thanks to the reviewer for his/her suggestion. Scale bars have been added in all microscope images in the new version of the manuscript.

In Figure 2A, the position for “$” label is wrong.

The “$” label in Figure 2A indicates a statistical difference between WT-HFD and KO-HFD mice, which is correct. As we have specified better the symbols in the new version of the manuscript, authors hope that now new symbols will ease the understanding of statistical differences between groups.

In Figure 3A and 3B, the data for F4/80 immunostaining are really unconvincing. The numbers for F4/80 staining-positive cells are too small to convince.

Thanks to the reviewer for his/her comments. The number of macrophages in the interstitial compartment has been calculated by counting the number of F4/80 positive cells in 6 microphotographs covering total renal cortex from each mouse. Therefore, the graph from Figure 3A is showing the mean value of F4/80+ cells per field.

On the other hand, we agree with the reviewer that the number of positive F4/80 cells found within the glomeruli shown in Figure 3B is unconvincing. The numbers were too low and we had difficulties to find representative images. For this reason, we decided to remove the information regarding glomerular F4/80+ cells localization from the manuscript.

The data for Figure 4A and 4B are really unconvincing. In lower panel of Figure 4A, the presentation of immunoblot for Gal-3 cannot be matched with densitometrical data plot (upper panel). The immunoblot showed no difference among WT-HFD, KO-SD, and KO-HFD groups, and there was a smaller Gal-3 protein expression for WT-SD. Why there is only one blot band for WT-SD group, but two repeat blot bands for other groups? In Figure 4B, the authors do not demonstrate that the positive Gal-3 stains are really located in endothelial cells. Moreover, from these IHC images, it is hard to distinguish the changes or differences among these four groups.

Thanks to the reviewer for his/her comments. We have chosen new immunoblots for Gal-3. In Figure 4A of the new version of the manuscript, new representative densitometrical data plots have been added. Currently, the bands shown coincide with one animal per group for clarity.

Regarding Figure 4B, we have added new representative images at lower magnifications as suggested by the first reviewer. Our objective was to describe Galectin-3 expression in the renal cortex and to correlate the expression of Galectin-3 with tubular lesion. However, for clarity we add a positive control suggested by the antibody manufacturer. Thus, we added in this letter an image for endothelial Galectin-3 staining in mouse pancreatic samples.

 (see the image on Word format document attached)

 The authors should show evidence to demonstrate that the endothelial Adam17 expression is really deleted in KO mice.

We would like to thank the reviewer for his/her suggestion. We have previously demonstrated endothelial Adam17 deletion in this animal model (PMID: 34073747) by immunohistochemistry performed in frozen samples of kidneys embedded in OCT. We were not able to localize Adam17 by immunohistochemistry in the present study as we did not collect frozen kidneys embedded in OCT. The immunohistochemistry of Adam17 performed in paraffin let to a huge background that difficult the interpretation of the results.

In Methods section, the descriptions for animal experiments (4.1.) are confusing. Why “proximal-tubular” Adam17KO male mice are used (line 252)? What is the source for WT and KO mice? Moreover, the authors measure the urinary ACR, but there are no descriptions for urine sampling, when and how?

We would like to thank the reviewer for his/her appreciations. We have corrected this typo in the new version of the manuscript (see line 286). We have also explained better the source of our animals (see lines 289). Regarding the origin of the urine samples, all were collected during the last week of the follow-up on morning spot urine collections through abdominal massage. In the new version of the manuscript has been described in more detail (see lines 322).

After detailed revision of the manuscript we detected a wrong graph in figure 2B. Data description and statistical analysis were correct, but the data presentation was wrong. We apologize for the error. In the new version of the manuscript the correct graph has been added.

Round 2

Reviewer 1 Report

The manuscript has been revised well. I think this manuscript is now acceptable.

Reviewer 2 Report

This revised manuscript can be accepted. No further comments.